# Review: Shear Properties and Various Mechanical Tests in the Interface Zone of Asphalt Layers

Hatim M. Akraym , Ratnasamy Muniandy *, Fauzan Mohd Jakarni  and Salihudin Hassim

Department of Civil Engineering, Faculty of Engineering, University Putra Malaysia, Serdang 43400, Selangor, Malaysia
* Correspondence: ratnas@upm.edu.my; Tel.: +60-03-8946-6373

**Abstract:** Over four decades, researchers have extensively focused on bonding flexible pavement layers. Scholars have concentrated on the partial or complete lack of interlayer bonding between asphalt layers, which is the primary cause of premature pavement failures, such as cracking, rutting, slippage of wearing courses, and decline in pavement life. These defects are observed within the high horizontal force areas owing to increased speed, braking, and sharp angles when entering or exiting highways and the variations in paving materials, traffic load, and climatic factors. Various studies have investigated the debonding of flexible pavements, and test methods have been developed to find effective solutions. This review is aimed at summarising and discussing certain factors influencing shear strength performance, such as tack coat material, surface characteristics of multi-layer construction of flexible pavements, and different mechanical shear tests. First, bonding in the interface zone area and its Effect on the shear strength performance is reviewed. Subsequently, the types of materials and construction methods and their effects on the bonding quality of the interface zone area are clarified. Finally, the linear relationships between certain effects and the Ability of nanofibers to improve the emulsion properties are discussed. However, no agreement on the optimum tack coat could be obtained owing to the variety of surfaces. Hence, a milling surface is recommended for higher shear strength. The shear test is the most used method for verifying the interlayer bonding strengths, and continuous research endeavours are recommended to analyse debonding in multi-layer asphalt pavements.

**Keywords:** interface bonding; tack coat; pavement mixture; surface texture; shear test

## 1. Introduction

Requirements on pavement performance in terms of Ability and durability bearing have steadily increased. Suitable pavement performance primarily depends on thickness and characteristics such as strength and stiffness of each layer. However, over the years, other factors, including the bonding between layers, have influenced the performance and lifetime of pavements [1]. To describe interface strength, bond adhesion may be sufficient. Nevertheless, in certain materials, the interface bonding in a double layer of flexible asphalt pavements cannot be characterised by adhesion alone as more complex stresses/strains and deflection must be transferred between the materials of the layers.

In the early 1970s, reported that the interface bonding between the layers of flexible pavements affected their performance by influencing the stress level of the materials. Additionally, the interface areas at these interfaces in the top layers of the pavements were strongly influenced by adhesion conditions. Uzan et al. [2]. Moreover, the Effect of low-grade interlayer bonding on the stress/strain distribution within a pavement design and the diminished Ability to support traffic could reduce the performance lifetime [3].

The Interface failure modes can be divided into sliding, pull-off, and tear failure. Sliding failure is the horizontal displacement between the top and lower layers of the interface under horizontal shear stress. Interface damage is related to sliding failure while

interpreting interlayer bonding accurately. Interlayer mechanical behaviour has been modelled using the elastic theory, Goodman, shear spring, multi-modal stress-strain curves, finite element model, influences of carrier layer stiffness, and Coulomb models are the commonly used models for the Evaluation of interlayer bonding conditions [4–6].

The elastic theory considers a thin layer with shear model modules G and thickness h. Due to pavement loading, the interlayer shear stress τ causes a relative shear displacement $\Delta\xi$ between the layers and is expressed as

$$\tau = G \cdot \frac{\Delta\xi}{h} = G \cdot \gamma \tag{1}$$

where γ represents the shear strain of the thin interface material. From Equation (1), the well-known Goodman's constitutive law can be obtained, which describes the interface behaviour in multi-layered elastic systems when the relative horizontal displacement of the double layers is $\Delta\mu$. The face of interlayer shear stress τ can then be summarised as

$$\tau = \kappa. \Delta\mu \tag{2}$$

where K = G/h (Mpa/mm$^3$) is the interlayer shear stiffness used to reduce numerical complications. For computer implementation, the following change stiffness variable can be used.

$$K = \frac{l}{1-l} 0 \leqslant l \leqslant 1 \tag{3}$$

where l is a shear stiffness parameter. For fully bonded and unbounded layers at the interface, Equations (4) and (5) are defined, respectively.

$$K = \infty, \quad l = 1, \text{ and } \Delta\xi = 0 \tag{4}$$

$$K = 0, \quad l = 0, \text{ and } \tau = 0 \tag{5}$$

The parameter K can be assumed as a characteristic value used to measure the level of interlayer bonding. However, when K is approximately zero, the interlayers tend to be fully sliding [7,8].

Based on the trend of dilatancy speed, a new interpretation of interface fatigue has been presented by theoretical models of the interface surface and fatigue progress [9]. Weimin Song et al. calculated the bonding fatigue performance between the open-graded friction course (OGFC) and the underlying layers by using the stiffness reduction method and the energy approach. They found a power law relationship between cumulative dissipated energy and fatigue life. The plateau value failure criterion appeared effective for evaluating the shear fatigue performance of multi-layer structures [10]. The interlayer shear stress varies synchronously with an increase in the relative displacement before failure. When the comparative displacement increases to a particular critical value, the shear stress reaches its highest value.

Based on the elastic layer theory, finite element method (FEM), and continuous finite layer method, several computer-based systems have been created to calculate the mechanical behaviour of asphalt pavement. Some examples of such software include the KENLAYER, BISAR, CIRCLY, EVERSTREES, EverStressFE, MICHPAVE, and 3D-Move Analysis [5,11–13]

The state of poor bonding is not fully understood due to several influences that differ according to the type of traffic load, paving materials, and climate. Moreover, an international agreement to standardise the type of test to measure interface shear strength (ISS) at the interface zone is lacking due to structural problems and material properties. Numerous investigations focused on poor bonding failure between pavement layers found several different influences, such as the type of asphalt in the wearing course [14], type of tack coat material, low compaction of base course, subbase course, or subgrade [15–17],

base course segregation, and poor or excessive tack coat application vehicle load [18–20], which affect the bonding between pavement layers.

Several laboratory and field studies have investigated several types of asphalt mixers and their surface textures. Liu et al. studied a composite structure of epoxy asphalt concrete (EAC) as a wearing course and stone mastic asphalt (SMA) on the upper surface to investigate the interface performance. They reported that the SMA+EAC composite structure, with the SMA13 mixture as the upper layer and epoxy resin-based bonding material, exhibited the most significant interface performance [14].

Research by the University of Oklahoma investigated the impact of new surface types of hot mix asphalt (HMA), aged and surface HMA, milled HMA, and Portland cement concrete (PCC). Additionally, the study confirmed the controlling Effect of surface texture conditions on the ISS, wherein the HMA specimens exhibited a significantly higher ISS than PCC [21]. Further, the National Cooperative Highway Research Program (NCHRP)Project 09-40 results using three different surface textures of sand mixtures, OGFC and SMA, reported a pronounced effect of texture [22].

The interlayer contact area is crucial to the interface bonding strength, and a higher texture depth increases the bonding strength and stiffness. Generally, a tack coat, fundamentally made from an asphalt emulsion or a small carbohydrate particle formed in an aqueous solution, is applied to increase the adhesion between various asphalt layers [1,23]. Several studies have extensively investigated the type and application rate of tack coat and have reported that these elements have a higher impact on ISS. An increase in the tack coat material rate caused a decrease in the interfacial bonding strength. In contrast, the shear strength was not significantly affected at relatively high temperatures in most cases [Yang, 2021 #1]. Shear testing or ISS is considered the most widely used evaluation method, as numerous conclusions have been drawn regarding ISS. Among the known shear tests, the two main types of tests are simple and direct shear tests. In addition, pull-off, flexural, tensile strength, cracker-resistance, and shear fatigue resistance tests have been recommended to analyse the bonding performance [24–26]. However, ISS is distressed by certain conditions such as temperature, cleanliness, interface friction, type of coat materials, and loading. Most of the existing test equipment is self-made and lacks standardised specifications in the laboratory or field. In addition, the test method and standardised specifications may not receive sufficient attention.

Thus far, the simple principle of interlayer bonding was introduced in terms of mechanical parameters and the influence of certain factors on the interlayer bonding performance was understood. This review aims to summarise and discuss some factors, such as the construction, applications, and test methods, which influence the shear strength performance. Further, the bond characteristics at the interface between layers are prone to be influenced by several factors. This review highlights some factors, including the type of tack coat, optimum application, surface characteristics, surface texture, surface condition, temperature, aggregate gradation, and compaction. Additionally, various shear test techniques in the interface zone on flexible pavements are discussed.

## 2. Materials and Additives Used for Tack Coat Layers

Emulsified asphalt has been recommended as a coating material [27]. However, cutback asphalt is not typically accepted in reverse coating applications owing to environmental concerns. Emulsified asphalt is a non-flammable liquid produced by combining asphalt and water with an emulsifying agent, such as soap or dust [28]. In this section, several types of tack coats are reviewed.

### 2.1. Asphalt Emulsion

Asphalt emulsions are water-continuous dispersions of fine asphalt droplets produced using a coiled mill. They are usually 1–10 mm in diameter with asphalt content of approximately 40–80% by weight [29].

The emulsions applied for coats include slow-setting (SS) grades SS-1, SS-1h, CSS-1, and CSS-1h; rapid-setting (RS) grades RS-1, RS-2, CRS-1, CRS-2; polymer-modified RS-2P; and latex-modified CRS-2L. Compared to cutback asphalt or hot asphalt binders, the wide usage of asphalt emulsions is promoted by the possibility of their easy application at lower temperatures, which saves more energy. Moreover, asphalt emulsions do not contain dangerous or volatile solvents; hence are non-flammable and pose no risks to users [29]. Other research summarised several studies on asphalt emulsions and concluded that asphalt emulsions, such as CRS-2, CRS-2P, CRS-2 L, NTSS-1HM, CSS-1, SS-1, SS-1h, and SS-1L, have higher interlayer shear strengths. Furthermore, various low-impact adhesive layers, such as CRS-1, were identified [19].

Different types of polymers can be used to prepare polymer-modified asphalt emulsions, such as ethylene-vinyl acetate (EVA), polyvinyl acetate (PVA), styrene-butadiene-styrene (SBS), styrene-butadiene rubber (SBR), epoxy resin, and natural rubber latex (NRL) [30]. A study by Qinqin et al. focused on the temperature performance of SBR-modified asphalt emulsions compared to a typical emulsion. A significant increase in the softening point was observed, corresponding with the heat stability and thermal resistance of asphalt. A temperature increase of 5 °C was observed in dactylitis, while the softening point showed a decrease, while the softening point showed a decrease [31].

Dianhao et al. investigated the performance of a modified asphalt that was created using trackless tack coat materials (TTCM). The study used SBS, uintaite mastic asphalt, masterbatch, and Sasobit wax modified with 50# base asphalt to obtain TTCM. The results showed that TTCM increased the interface shear strength (ISS) by 69% at 20 °C. TTCM can function as a replacement and prospective candidate in high-temperature pavement applications compared to conventional tack coats [32]. Hence, a polymer-modified asphalt emulsion can seal the base layer when producing a high binder content near the interface with high application rates [18].

Fibres are one of the most common additives in asphalt and exist as polypropylene (PP), polyacrylonitrile (PAN), lignin, and basalt fibres. Du. Studied the Effect of different fibres on the performance of cold recycled asphalt emulsion mixed with PP, PAN, lignin, and basalt fibres. The results suggested that the polyester fibres significantly improved the fatigue life compared to other fibres [33]. Therefore, polyester fibres are the best-recommended option for improving the performance of recycled emulsion mixtures [34,35].

Feipeng et al. used an SBS-modified asphalt binder to produce an asphalt emulsion that satisfied the specifications typically required for micro-surfacing production. Additionally, three nanofibers produced from sisal, tree, and cotton, denoted as MasterSeal NP 1, 2, and 3, respectively, were employed. Further, the asphalt emulsion was modified using 0.5% by weight of asphalt residue of the nanofibers. The study's findings showed that nanofibers generally improved the durability and rheological properties of the resulting material [36].

*2.2. Hot Asphalt Cement*

Emulsified asphalt has been reported to exhibit more interfacial bonding than hot asphalt cement [19]. Conversely, asphalt cement must be adequately heated before spraying; else, uniform coating of asphalt cement on the surface layer of pavements at low application rates is difficult. In Georgia, hot asphalt cement, such as AC 20 and AC 30, is routinely used as a tack coat [18]. Asphalt cement, including PG 64-22, PG 67-22, and PG 76-22 M, are used as tack coats as they exhibit excellent performance while bonding layers of asphalt pavements. However, they are not environmentally friendly and pose operational difficulties [19]. Table 1 presents recommended tack coat application rate based on the surface type.

**Table 1.** Recommended tack coat application rate based on the type of surface [37].

| Pavement Condition | Application Rate (gal/yd$^2$) | | |
|---|---|---|---|
| | Residual | Undiluted | Diluted (1:1) |
| New HMA | 0.03~0.04 | 0.05~0.07 | 0.10~0.13 |
| Oxidised HMA | 0.04~0.06 | 0.07~0.10 | 0.13~0.20 |
| Milled HMA | 0.06~0.08 | 0.10~0.13 | 0.20~0.27 |
| Milled PCC | 0.06~0.08 | 0.10~0.13 | 0.20~0.27 |
| PCC | 0.04~0.06 | 0.07~0.10 | 0.13~0.20 |

Xiaoyang Jia et al. reviewed the breakdown of tack coats in an orthotropic steel bridge deck overlay, where two tack coats, hot-melt and solvent-borne coating, were used; a solvent-borne coating is a type of polymer-created cement capable of flowing at average temperatures. The analysis indicated that the tack coat materials significantly reduced the strain at the base of the overlay compared to the stress at the tack coat top. Therefore, the hot-melt coating was recommended as a better material for tack coats [38].

*2.3. Cutback Asphalt*

The usage of cutback asphalt was limited by the evaporation of its volatile chemical content associated with environmental issues. Furthermore, considerable energy is required while manufacturing cutback asphalt from petroleum solvents, which makes it more expensive than emulsified asphalt, where water and other emulsifying agents are used. Ghaly et al. evaluated the influence of cutback asphalt grade 60/70 and latex-modified tack coat asphalt emulsion. The results indicated a higher ISS of the modified tack coat than that of the cutback and tack coat asphalt emulsion. In addition, a slight improvement in the shear strength was observed at low viscosity compared to high viscosity [39].

Kulkarni et al. evaluated pyrolysis oil from the pyrolysis of low-density polyethene (LDPE) waste to realise efficient, inexpensive, and environmentally friendly bitumen modification. Pyro-oil was added to plastic waste-derived bitumen VG 10 and investigated for use as a substitute for diesel during the preparation of modified cutback bitumen; during the preparation of modified cutback bitumen, pyro-oil was added to plastic waste-derived bitumen VG 10 and investigated for use as a substitute for diesel, which serves as a tack coat between the two layers of bituminous pavement. The results showed that the plastic waste-derived pyro-oil at an added percentage of 20% was ideal for the preparation of modified cutback; it was found to be a good tack coat. Further, improvements in the shear strength value following the addition of pyro-oil at a rate of 0.20 kg/m$^2$ were observed. The study recommended a comparison between the performance of pyro-oil–modified cutbacks with those modified with diesel, kerosene, and petrol [40].

*2.4. Application Rate of Tack Coats*

Over time, the application rate of tack coat materials has been extensively studied. Several investigations have shown the effects of application rate and the improvements to the ISS results. Moreover, the application rate increases the contact area and shear strength. Thus, if the adhesive layer is excessively heavy, it can introduce a sliding level at the interlayer and reduce the bonding. Therefore, an excellent adhesive application rate is essential for achieving a high ISS between pavement layers. Notably, no agreement exists on tack coat quantity and application rate [40]. The overall residual rates lie in the range of 0.03–0.20 gal/yd$^2$ for different types of pavements, as presented in Table 1 [37].

To define the optimal tack coat application rate, Ghaly et al. evaluated the practice of tack coat application through simple laboratory shear tests. The results showed improved bond strength as a function of application rate, temperature, and viscosity. Further, the shear resistance increased significantly at the interface with an increase in the application rate; however, it decreased with an increase in the temperature. Similarly, the bond strength of the emulsion improved slightly at lower viscosities compared to higher viscosities. Asphalt emulsion was applied at an application rate of 0.25 L/m$^2$. The inter-surface

bonding strength increased with the application rate (average 1.3 and 1.47 Mpa to reduce asphalt and asphalt coating emulsions, respectively). The bonding strength between the layers was reduced at higher application rates of 0.35 L/m$^2$ [39].

Varanine, S. et al. studied the relationship between ISS and the application rate. The highest value of ISS was realised following the application of a 0.6 l/m$^2$ TTCM-type thinner cladding layer at the rate of 60:40 on a newly prepared HMA surface. The same application rate was applied to the unmailed surfaces, resulting in a higher ISS. The research recommended an application rate of 60:40 for 0.25 to 0.35 L/ m$^2$ of TTCM, which can be increased to achieve better performance [37]. However, variations in the application rates exist based on the type and age of the surface and the difference in the texture depth [37]. Reviews of existing studies have stated that the application rates range from 0.03 to 0.08 gal/yd$^2$ depending on the time and type of asphalt mixes. Table 2 lists the optimum application rates of certain types of emulsions, such as CRS-2P, CSS-1h, and SS-h, for different bottom layers [37].

**Table 2.** Optimum application rates of CRS-2P, CSS-1h, and SS-h residue for different bottom layers [37].

| Tack Coat Type | Optimum Residual Application Rate (L/m$^2$) | | | |
|---|---|---|---|---|
| | New HMA | Aged and Worn HMA | Milled HMA | Grooved PCC |
| CRS-2P | 0.7020 | 0.7020 | 0.1400 | 0.1400 |
| CSS-1h | 0.4100 | 0.1400 | 0.1400 | 0.2810 |
| SS-h | 0.2810 | 0.2810 | 0.1400 | 0.1400 |

Rouzbeh et al. studied the Effect of tack coat emulsion type and application rate on early-aged interlayer shear strength of pavements in cold regions. An objective of the study was to determine the optimum application rates of tack coats, such as CRS-2P, CSS-1h, and SS-h, which are widely used in cold climates. As shown in Table 2, low and intermediate applications of CSS-1h tack coat (0.140 and 0.281 L/m$^2$, respectively) improved the ISS, while the applications of the SS-h tack coat on the newly prepared HMA surfaces at high and intermediate rates (0.702 L/m$^2$ and 0.281 L/m$^2$, respectively) significantly increased the ISS and PCC surfaces. Furthermore, applying CSS-1h at a lower rate (0.140 L/m$^2$) improved the ISS value [41]. Table 3 summarises classifications and comparisons of several types of tack coats. An optimum tack coat should satisfy the surface and performance conditions. Thus, tack coat rates have been investigated extensively. However, no uniform conclusion on whether the optimum tack coat rate exists under specific application conditions has been drawn.

**Table 3.** Summary of Classifications and comparisons of various tack coat types.

| Tack Coat Types | Classification | Features | References |
|---|---|---|---|
| Asphalt Emulsion | Slow-setting (SS): SS-1, SS-1h, CSS-1, and CSS-1h Rapid-setting (RS): (RS-1, RS-2, CRS-1, CRS-2, and CRS-2P | Easy handling, energy saving, environmentally friendly, and personnel safety | [29,37,42] |
| | Polymer-modified: CRS-2P, Trackless Tack Coat, Latex-modified: SS-1h, CRS-2L Polymères types: EVA, PVA, SBS, SBR Latex, EPOXY resin, and natural rubber | High bond strength, environment friendly, solve existing issues, energy saving | [19,30,31,43] |
| | Fibre modified: PP, PAN, Basalt Fiber, and Nanofiber | High bond strength, environment friendly, solve some existing problems, energy saving | |
| Hot Asphalt Cement | AC-20 and AC-30, PG 64-22, PG 7622, and PG 58 | High bond strength, difficult to spray, high heating | [10,37,38] |
| Cutback Asphalt | RC 70, asphalt grade 60/70, VG 10 with pyro-oil | It poses environmental problems, requires more energy, and is expensive | [23,39,40] |

### 3. Aggregates and Surface Textures

Pavement texture refers to the surface irregularities of pavement in contrast to a planned surface. Surface texture can be categorised into micro-texture, macro-texture, or roughness. Each of these categories has a specific wavelength range and plays a specific role in tire-pavement interactions. The macro texture is influenced by several factors, including aggregate size (fine or coarse), air voids, binder content, and a viscosity [37]. Pavement friction is significantly influenced by the amount of aggregate in the mixture because aggregates account for approximately 95% by weight and 90% of the volume of most asphalt mixtures.

Generally, macro-texture depends on the ratio and characteristics of the coarse aggregates used; therefore, the mix designer is expected to consider these criteria when selecting proper aggregate grades for design purposes [44]. Numerous studies have acknowledged that the ISS of milled pavement surfaces is always higher than that of non-milled pavement surfaces [37]. Moreover, texture depth and surface roughness are known to be strongly related to the properties of the materials [10].

For surface texture measurements, several methods have been designed, such as the ASTM E965 sand patch method, which was developed to determine the mean texture depth (MTD) of pavement surfaces. A pre-known glass sphere volume on the surface was considered to make a pancake for computing MTD by measuring the area of the formed pancake and dividing the obtained value by a known volume [45].

Measuring macro-texture is the most commonly used technique for determining pavement texture. A majority of the conventional methods rely on touch measurements. Conversely, other techniques typically rely on non-contact methods, including laser and photometric stereo techniques and low-speed friction measurement devices, such as the British portable tester. In addition, electro-optic and laser-based techniques can be used for measuring high-speed friction. A more contemporary laser-based CTMeter, and laser texture scanner (LTS) has been developed by Ames Engineering. Dynamic friction tester (ASTM E1911, 2002) and outflow meter outflow time (OFT) ASTM E2380-05 have also been developed [46]

Typically, a tack coat is applied to existing surfaces to determine the adhesive bond between the surfaces of existing pavement and newly laid asphalt. Therefore, the characteristics of tack coat and bonding strength are significantly affected by the type of asphalt mix and the upper and lower face textures [18]. Along with the top surface texture, which maintains an appropriate safety level for road vehicles, the bottom surface textures also contribute to the interface between the layers of asphalt and pavement. The interface is influenced by the type of emulsion used as the tack coat between the pavement layers [44].

Furthermore, shear strength differs significantly at low and medium application rates. However, only slight differences are observed at high application rates due to the possible lower contribution of the effects of the microstructural features on the surface roughness or texture when filled with tack coat materials [28].

Weimin et al. investigated the factors affecting the shear strength between OGFC and the base layer, primarily using laboratory tests. A type of SMA and two types of dense-graded surface mixtures were used in the study. In addition, the interface friction effect on the shear strength was investigated by determining the surface texture depth of the base layers. The results indicated that the influence of the surface texture depth of the base layer on the ISS was affected by the tack coat application rate and vice versa. Conversely, other factors were not significant at low tack coat rates or low texture depths, with the surface texture depth of the base layer indicating the roughness of the interface; these parameters were found to be well correlated with shear strength [47].

Julián et al. determined the amount of bituminous tack coat and its contribution to the texture of the milled asphalt layers. Their study aimed to reveal the impact of an increase in the surface area of milled pavement on the amount of tack coat. They found that the texture created by the milling process was similar to that found in the sand patch test. Grooves facilitating bitumen emulsion runoff were created, which also promoted

excessive deposition of bitumen in their valleys due to excess dosages [48]. The surface texture of the pavement had repetitive features and self-similarity, representing fractal characteristics. Fractal geometry provides a suitable method for describing the irregularities and complexities of pavement surface textures.

Hou et al. studied the quantitative relationship between area fractal dimension and gradation, which are used to correlate aggregate gradation and the British pendulum number (BPN). Further, the calculation formula of the area fractal dimension of aggregate distribution characteristics were deduced based on their proposal of the mass fractal characteristic function. Moreover, the surface textures of five different types of asphalt mixes: AC, micro-surfacing MS, rubber AC RAC, OGFC, and SMA, were considered. The results of this method, as shown in Figure 1. include the measurements of pavement surface roughness [49].

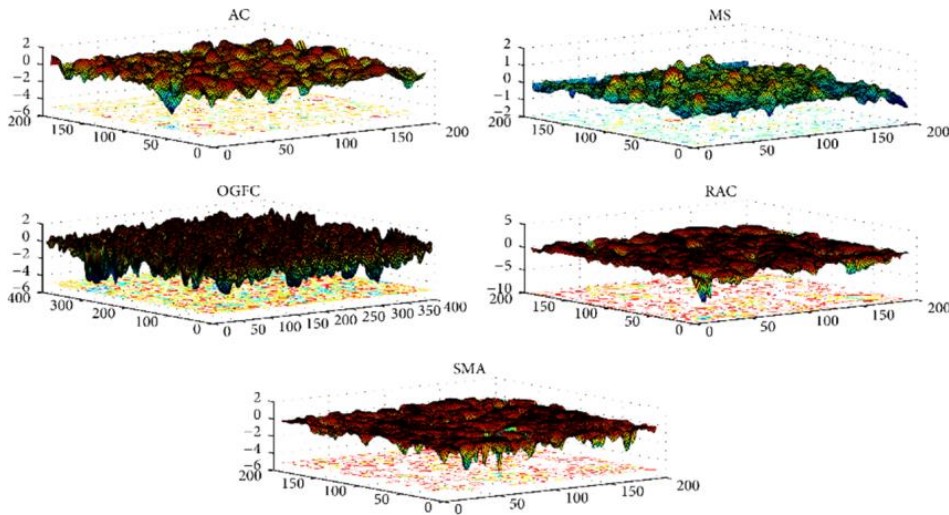

**Figure 1.** Surface textures of five different pavement types.

### 3.1. Surface Conditions

Bond strength at the interface is affected by pavement surface conditions such as texture, cleanliness, and wetness. Guidelines on the required surface condition of existing pavements before tack coat application have been specified in several handbooks. These guidelines recommend the application of a tack coat only on clean and dry surfaces. A summary of the surface preparation and weather conditions for proper tack coat application was provided by the Asphalt Institute [37]. The Asphalt Institute manual with series no. 22 on the construction of HMA pavements recommends the application of a tack coat under the same weather conditions as HMA paving. In addition, the manual suggests that the surface must be dry and clean before applying the tack coat [50]. The Basic Asphalt Emulsion Manual, manual series no. 19 (MS-19) of the Asphalt Institute, also recommends the application of tack coat only on clean and dry surfaces [51].

Mazumder et al. reviewed the installation and implementation of a proper tack coat application and found that no limit to the moisture content that may be present during tack coat application was specified. However, excessive water affects the shear strength of tack coats, particularly during construction. Rainwater can significantly reduce the interlayer shear strength of tack coat materials [52]. Several factors affect the performance of tack coat layers, particularly during construction. Therefore, the impact of moisture content on the damage prevention mechanism of tack coat bonds in pavements has been studied under dynamic truck loads. A study found that the presence of dust contributed least to shear damage, while the occurrence of rain during the process significantly contributed to the damage mechanism. Further, the application of a tack coat on a wet surface increased the damage potential by 20.1%. In addition, a 12.8% higher damage level was observed upon an approximately 50% tack coat coverage application during construction compared to

that for a 100% tack coat coverage of the surface area [53]. Seoe et al. studied the bonding potential of trackless tack coats and found that the bonding strength of the surface types played a more critical role in determining the shear bond strength [54].

### 3.2. Aggregate Gradation

The shear strength of the interlayers originates from the interlock formed upon the penetration of the aggregates of one layer into the voids of other layers. Paving the upper and lower layers with dense-graded mixtures results in adequate adhesion due to extending five contact areas between the interlayer surfaces. Aggregate gradations of bonding mixtures are considered to contribute significantly to the level of the achieved shear strength; fine and coarse-graded mixes affect bond strengths differently [17,19,55–57].

The NCAT report 05-08 deduced that the type of asphalt mixture is a significant factor for bond strength. The analysis described the impact of fine-graded smaller (NMAS) mixtures on bond strength, which was more than that of coarse-graded (NMAS) mixtures. However, the texture of the mixture type significantly impacts the outcome [58].

Chen, J.-S et al. investigated the influence of surface characteristics on the bonding properties of bituminous tack coats by evaluating the interlayer characteristics of three different asphalt mixtures: DGAC, SMA, and PAC, as shown in Figure 2. The nominal size of the mixture at the top was 19 mm. PAC is an open-graded mixture, while SMA is a gap-graded mixture with 9.5- and 3.2-mm deficiency in size, respectively. Since PAC and SMA are coarse mixes, the aggregate gradation of the three mixtures was designated as dense-graded asphalt. However, the PAC mix contained fewer fine aggregates than the SMA mix, thereby rendering it more permeable with approximately 20% air voids compared to only 4% in SMA. The test planned two emulsions, three surfacing types, three test temperatures, four stresses, and six residual emulsion rates. The study concluded that the bond properties at the interlayer could be determined from the surface characteristics [59].

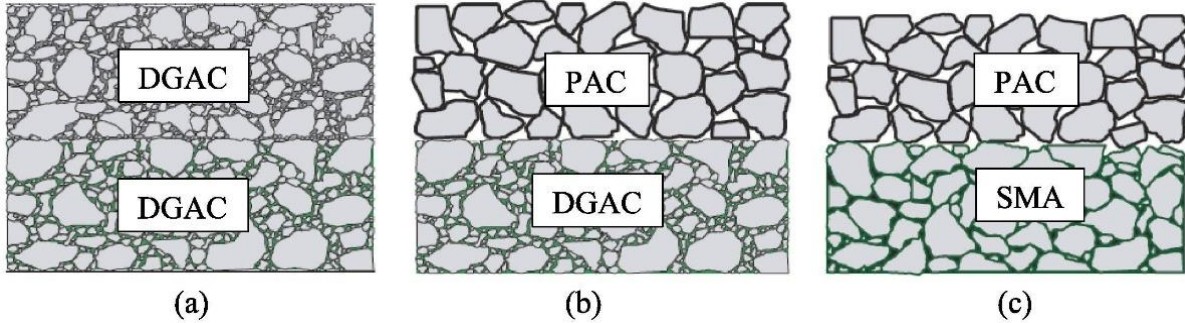

**Figure 2.** Different asphalt mixtures, including (**a**) DGAC-DGAC, (**b**) PAC-DGAC, and (**c**) PAC-SMA [59].

Various researchers agree with the study of MTD for higher shear strength using milled surfaces. However, certain studies highlight the necessity of 3-D image processing while studying pavement surface layers. Moreover, emulsions have to be applied on clean and dry surfaces, and a large amount of water can reduce the shear strength. Furthermore, aggregate gradation is essential for interlocking layers with a low percentage of air voids. Table 4 illustrates a summary of the surface textures, surface conditions, and aggregate gradation discussed thus far.

**Table 4.** Summary of the discussion on surface texture, surface conditions, and aggregate gradation.

| Factors | Recommendation | References |
| --- | --- | --- |
| Texture characteristics | Agreement with a milled surface for higher shear strength. Traditional method: MTD. New method: 3-D image processing. | Chen, et al. [30]. Mohammad et al. [23]. Zhang, W.et al. [10], Wang, J. et al. [9]. Miao, Y. et al. [32] |
| Surface texture | Top texture: important for vehicular safety. Bottom texture: important for interface pavement layers and affected by the emulsion applied. | Khasawneh, M.A. and M.A. Alsheyab [44,47,48]. Manual, et al. [51] |
| Surface conditions | The emulsion should be applied on a dry and clean surface, the limit of moisture content should be specified, and a large amount of water can reduce the shear strength. | Song, et al. [47], Varamini, S., et al. [60] Estaji, et al. [53], Manual, et al. [51] |
| Aggregate gradation | Aggregate gradations are necessary for layer interlocking, an increase in MTD. The interlock is enhanced, while the shear strength is reduced when the mixture is designed with a high percentage of air voids. | West, et al. [58], You, L., et al., [57]. Kruntcheva, et al. [55] |

## 4. Temperature

Temperature is an essential factor affecting asphalt behaviour, as either an increase or decrease in temperature affects the characteristics of the asphalt binder and tack coat. Thus, a linear relationship exists between temperature and ISS, where an increase in temperature causes a decrease in the ISS [19,61,62]. The factors that affect the properties of the bond between different layers of asphalt pavement, including temperature, have been studied extensively owing to their Effect on ISS. The difference in temperature between the wearing and binder course during the construction processes should be considered, as the probability of inter-layer slippage may increase [1]. The NCHRP report 712 stated that an increase in temperature from 10 to 60 °C caused a significant increase in ISS. Additionally, the bonding performance of the trackless emulsion was evaluated at temperatures greater than 40 °C, as measured using ISS, and the results were better than that of the CRS-1 emulsion [63].

Another study evaluated the bond strength between layers of pavement, and the results showed that temperature was among the most significant factors affecting the bond strength. The observed bond strengths at 50 °F were 2.3 times higher than those at 77 °F [58]. Furthermore, the factors affecting the shear strength between OGFC, and the underlying layer were studied, and the results showed that the temperature had the greatest impact on the ISS of the OGFC and SMA composite considered with or without a tack coat. An increase in the temperature from 0 to 50 °C drastically reduced the ISS from approximately 1000 to 100 kPa, possibly owing to the dramatic change in the viscoelastic nature of the stiff asphalt material from soft to hard upon an increase in the temperature from 0 to 50 °C [47].

Ahmed et al. investigated the factors affecting the ISS of field and laboratory samples. The research showed that the significant challenge faced in using samples prepared in the laboratory for the prediction of the field behaviour of tack coats was caused by the conditioning temperature of the samples in the laboratory before testing in the field. The test temperature had a pronounced effect on the surface texture, particularly at higher temperatures, as shown in Figure 3. Thus, the Effect of testing temperature on the ISS was significantly higher at elevated temperatures, thereby causing the coring of both the field and laboratory samples.

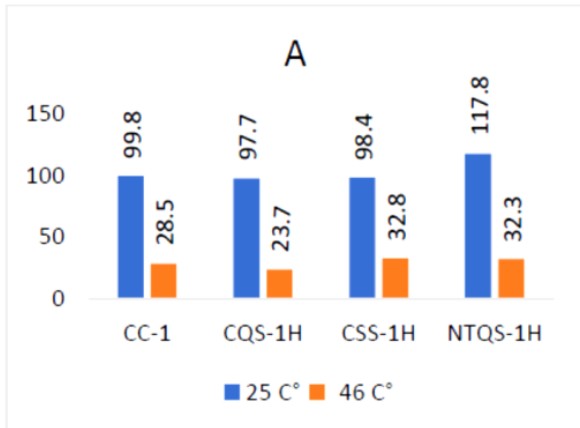
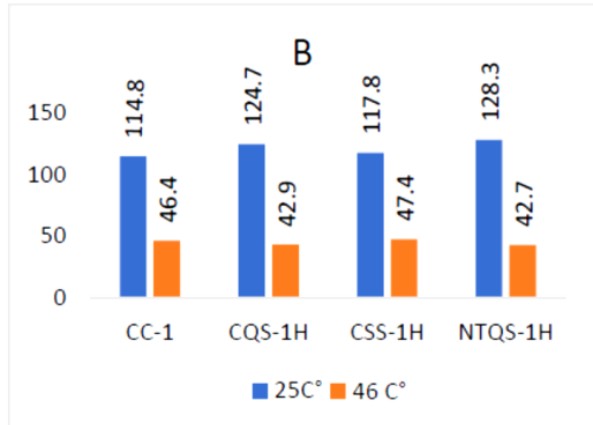

**Figure 3.** Effect of test temperature on ISS using (**A**) low surface textures for the bottom layer (0.05 gal/yd$^2$) and (**B**) high surface textures for the bottom layer (0.05 gal/yd$^2$).

## 5. Compaction Procedure

The shear failure of multi-layer asphalt pavements typically occurs at the interface between the top and lower asphalt layers through asymmetric stiffness. The factors that promote good bonding between the multiple layers of a pavement structure are the construction temperature and compaction level of the upper and lower layers. The performance of asphalt pavements can be predicted using laboratory samples; therefore, consistent compaction of laboratory samples and a good correlation with the compaction of field samples are essential [64].

Various laboratory compaction techniques are available, such as gyratory, impact, rolling wheel, and kneading compactions. Impact compaction, which is the Marshall compaction method that is still being used, is the first method that was proposed [64,65]. The asphalt mix design can be performed using the Marshall and SGC methods. In the Marshall method, volumetric parameters are considered, whereas in certain cases, the considered parameters are the Marshall stability and flow. The first level of the Superpave method considers volumetric parameters before including mechanical properties, such as tensile strength, stiffness modulus, and fatigue resistance. Consequently, different laboratory compaction methods can yield specimens with varying volumetric characteristics for the same asphalt content when other parameters are kept fixed. Scholars have considered compaction by rolling as the best method for field compaction simulations [66].

Other compaction devices currently available include static and vibratory compactors, rolling wheel actors, and French plate actors. Recently, studies have evaluated the significant reasons for the observed variations in field and laboratory compaction to understand the impact of the selected compaction method on the mechanical performance of the materials [17,67,68]. The factors influencing the interlayer shear strength of the field and laboratory samples were studied, and the results showed significant variations in their ISS owing to differences in the adopted compaction method. Figure 4. shows the coring Effect on the ISS of the laboratory-prepared and cored samples. Since the gyratory compaction method is primarily used in laboratories, significant differences in the pore structure at the interface layer were observed in the laboratory when compared to that in the field [67].

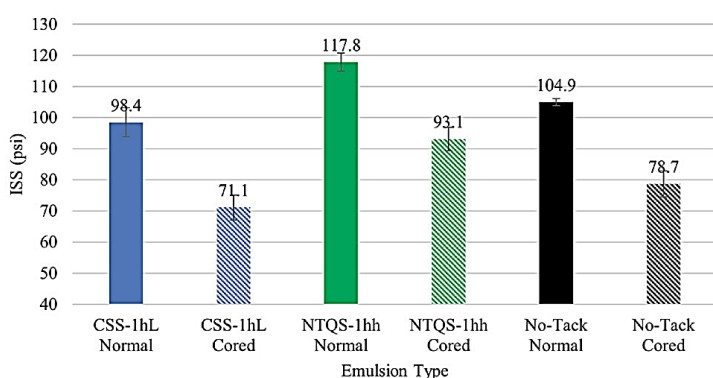

**Figure 4.** Effect of coring on ISS of laboratory-prepared and cored samples.

Owing to this difference between the field and laboratory compactions, researchers at the University of Putra, Malaysia, under the Ministry of Science and Technology, created a rotary compactor for the uniform compaction of SMA asphalt mixtures to the required slab density and thickness. The method requires the use of a simulative rotary compactor to prepare heavy-duty laboratory asphalt mixtures. The rotary compactor can produce asphalt mixtures with the required density, air voids, Marshall flow/stability, and resilient modulus [69].

SMA mixtures require a particular compaction system to reach the desired proportion of air voids and other mechanical properties. Muiandy et al. developed Turamesin, a laboratory slab compactor, for the compaction of asphalt mixtures in the laboratory using field simulation conditions. The results were promising and achieved an optimum air–void ratio. However, the impact of compaction techniques on the internal structure of the slab was recommended for further studies. Table 5 summarises the discussion on temperature and compaction conditions.

**Table 5.** Summary of the discussion of temperature and compaction procedure.

| Influencer | Recommendation | References |
|---|---|---|
| Temperature | A key factor, if increased or decreased, changes the shear strength. | Sufian, A.A., et al. [53], Aire, G., et al. [70], Muniandy, R., et al. [68] |
| | When used for analysing ISS from 10° to 60 °C, the ISS of the trackless emulsion was higher than that of the CRS-1 emulsion | Al-Qadi, I.L., et al. [63] |
| | At 10 to 15 °C, epoxy binder improved the interlayer bonding | Apostolidis, Liu [71] |
| | In OGFC-SMA mixtures, with the increase in temperature from 0 to 50 °C, the shear strength drastically decreased from 1000 to 100 kPa | Weimin Song et al. [47] |
| Compaction | Mostly Marshall and SGC methods are widely used. | Leandro, et al. [52]. Sufian, A.A., et al. [53], Airey, G., et al. [55] |
| | Differences between laboratory and field compaction are discussed. | Sufian, A.A., et al. [53]. Muniandy, R., et al. [59] |
| | Laboratory roller compactor and vibrating roller gave similar qualities of interlayer bonding. | |
| | The rotary compactor and wheel tracker (RCWT) developed by the University of Putra, Malaysia, can produce asphalt mixtures with the requirements of air void, density, resilient modulus, Marshall stability and flow. | Muniandy, R., et al. [53]. Moazami, D. and R. Muniandy [23] |
| | Turamesin, a laboratory slab compactor, is recommended for stone mastic asphalt (SMA) mixtures. | Muniandy, R., et al. [68] |
| | Gyratory compactor (gyration angle of 1.25°) is the best approximate compaction in fields | Reba's, C.Y. and L.P. [16] |

### 6. Different Mechanical Shear Tests in the Interface Zone

Recently, several countries have developed improved methods for testing interlayer bonds in asphalt layers. However, no internationally recognised methods are available at present for testing purposes because the results from different laboratories are always different in numerous aspects.

Thus, laboratory studies on the estimation of the bonding between adjacent layers of pavement have increased; numerous test methods for the characterisation of interface bonding have also been recommended [72]. Currently, certain methods, such as destructive interlayer test methods, require laboratory testing of at least a couple of layers from a pavement structure. Such specimens are brought to failure at constant loading conditions based on the test device employed. The significant disadvantages of this method include the time required and destruction caused to certain parts of the pavement, thereby causing traffic delays. Based on test configurations or loading procedures, the destructive interlayer test methods are grouped into four sections torque, tensile, wedge splitting, and shear tests (with or without average load), as shown in Figure 5.

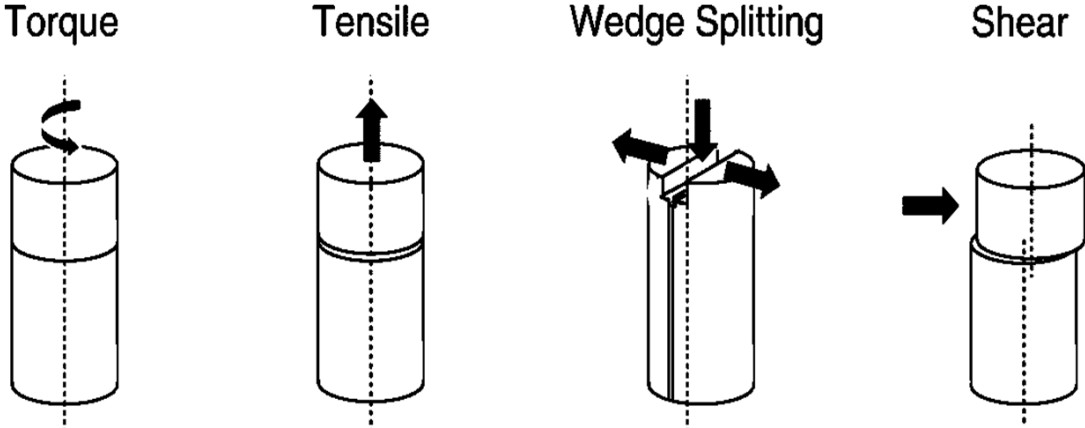

**Figure 5.** Testing of interlayer bonding in asphalt pavements.

#### 6.1. Torque Test

The interlayer torque test aims to cover the peak shearing torque applied to cored specimens to debone the surface system d from its substrate. The top of the specimen is subjected to torque to induce twisting failure in the bond layer. In the UK, the test is performed using a handheld torque wrench and can be performed on specimens with diameters of 100 and 150 mm [73]

A new field torque test method (OFTT) was developed by Mahmoud et al. for the Evaluation of in situ tack coat performance using a cost-efficient, smaller destructive prototype field test device. The study evaluated the shear tests of cores acquired from the field to constrain the effectiveness of the OFTT and found that the calculated peak torque values by the OFTT showed a high correlation with the laboratory-calculated shear strength [74]. Further, the value of the torque momentum at failure M (Nm) was recorded, and the bond torque resistance τ (in MPa) was calculated using Equation (6).

$$\tau = \frac{12M \cdot 10^3}{\pi D^3} \tag{6}$$

The performance of a laboratory-created automatic torque-bond test was demonstrated by Collop et al. The device performed both quasi-static and repeated load interface tests and was developed for servo-hydraulic testing machines as it could apply tensile or compressive force under different loading conditions (static or cyclic). The machine was operated using a Rubicon digital servo control system. Consequently, the signals from the load cell and LVDT were continuously monitored and recorded using the Rubicon data acquisition software.

The outcome of the repeated load (fatigue) tests showed that at low temperatures, the fatigue life was higher from the shear stress perspective, with a higher level of sensitivity to the shear stress level [75]. Table 6 presents different torque-test methods available.

**Table 6.** Different torque test methods.

| Categories | Device Name | Reference |
|---|---|---|
| Torque method | United Kingdom torque test<br>Oregon Filed Torque Test (OFTT)<br>Automatic Torque bond Tester<br>Shear torque fatigue test<br>ATACKERTM method | Canestrari, F., et al. [73].<br>Mahmoud, A., et al. [74].<br>Collop, A., et al. [75].<br>Ragni, D., et al. [76].<br>Wang, J., et al. [18]. |

Davide et al. studied the shear–torque fatigue performance of geogrid-reinforced asphalt interlayers for different test purposes. The method provided practical information on the condition of interlayer bonding and interlayer fatigue failure of reinforced systems under stresses and strains [76].

In addition, the ATACKERTM method can be employed to determine torque strength in situ and in the laboratory. The method involves a similar test procedure and can be applied in place of vertical tensile loads [18]. Laboratory studies have determined several difficulties that are still encountered in traditional torque bond tests, such as being only suitable for field use, applicable only to the top-most interface of the pavement, fixed specimen diameter, incorrect torque rate owing to manual operation, and the occurrence of axial bending [77].

### 6.2. Tensile Tests (Pull-Off Tests)

Pull-off tests can be performed both in the field and laboratory. In the test, a double-layered specimen is subjected to tensile stress, with the top surface layer pulled up axially to break the interface. However, the essential drawbacks of this test include the unfeasibility at higher interlayer bonding resistance, particularly when it is greater than the tensile resistance of the materials, and the difficulty in assessing the produced grain interlock during compaction [73]. Various scholars have used tensile tests to evaluate the performance of bonding layers [18,78–81]. For instance, Louay et al. proposed the Louisiana tack coat quality tester (LTCQT) for the field evaluation of the bond strength of tack coats. The LTCQT can determine the bond strength quality of tack coats and successfully distinguish between the responses of the studied tack coats. However, the softening point was identified as an essential parameter for determining the optimum temperature for the tack coats [78].

Abdur-Rahim et al. developed the compression pull-off test (CPOT) as a new method for evaluating the bond strength of bitumen and mastics. The CPOT successfully determined the adhesive and cohesive bond strengths of the binder. In addition, CPOT addresses a few problems encountered while using the standard test methods [79]. The use of a pneumatic adhesion test to identify the moisture sensitivity of asphalt binders has also been recommended [18]. This method was developed at the National Institute of Standards and Technology (NIST) for the testing of coatings and has now become an aspect of ASTM D 4541, which is referred to as a pull-off strength coating using a portable adhesion tester.

The University of Texas at El Paso (UTEP) developed a pull-off test to measure the tensile strength of the tack coat before paving a new overlay. The method could measure the tack coat strength under the tension mode instead of the shear mode [82]. Hakimzadeh et al. also proposed an interface bond test (IBT) as a simple method for testing specimens fabricated from either field cores or cylindrical samples produced in the laboratory. In addition, IBT could determine the bond properties of thin layers, such as thin-bonded overlays. Further, computational models can rely on the tensile fracture data from the IBT to optimise systems; the data can serve as a link between the properties of materials and performance in the field [83]. Table 7 lists the different tensile tests and methods.

**Table 7.** Different tensile tests and methods.

| Categories | Device Name | Reference |
|---|---|---|
| Tensile test (pull-off) | Louisiana tack coat quality (LTCQT.) | Rahman, A., et al. [78] |
| | Compression pull-off test | Mohammad, L.N., et al. [79] |
| | Portable adhesion tester | Wang, J. et al. [9] |
| | UTEP pull-off devices | Zhou, L., et al. [82] |
| | Interface bond test | Hakimzadeh, S., et al. [83] |

*6.3. Wedge Splitting Tests*

Wedge-splitting tests involve pushing a slender wedge into the interface of a bilayer sample to ensure the separation of layers via the horizontal component of the applied force [77]. The test requires a wedge that advances into a double-layered sample at a rate of 2 mm/min, and the result is obtained as a force–time curve [73]. Elmar et al. proposed a simple method for measuring the fracture-mechanical behaviour of layer bonds based on simple tests performed on inter-layer bonds. This new wedge-splitting method produced valuable results and could be applied to both prismatic and cylindrical specimens; however, quantitative bonding characterisation was not possible [84]. Jamaaoui et al. developed a new test method using wedge-splitting tests. The test relied on optical measurements of bi-layered asphalt concrete with and without a carbon fibre grid. Optical measurements enabled the calculation of the crack relative displacement factor, while an analytical approach was used to determine the stress intensity factor. In addition, a mark-tracking method was used to measure the crack opening displacement [85].

*6.4. Interface Shear Strength Test*

While no international agreement exists on a test method to determine the ISS between pavement layers, the shearing method is a suitable mechanism for studying the state of interface bonding and is the most widely used test for investigating interface problems owing to its smooth operation and suitability. The interfaces of the double-layered specimens are subjected to a constant rate of shear displacement/shear loading until failure. The shear stress can be computed using Equation (7).

$$\tau = \frac{4 \times F}{\pi d^2} \tag{7}$$

where $\tau$ is the shear stress (MPa), F is the shear force (N), and d is the diameter of the specimen (mm). The outcome of this test is primarily shear strength [73]. Various countries have developed several types of shear test equipment, which can be classified into pure direct shear tests (without normal stress) and direct shear tests (with normal stress applied).

6.4.1. Pure Direct Shear Tests

Certain simple tests have been widely employed in testing shear strength [86]. In 1978, Leutner developed primary interlayer direct shear devices for use in shear tests. The test necessitated the application of shear strength at a constant rate through a predefined plane until displacement. Further, the ongoing monitoring of the resulting shear force was considered a function of the applied displacement [2,73,77]. However, the Leutner test has been extensively modified to suit different purposes, such as to improve compatibility with the FDOT test device, Laboratory Caminos Barcelona (LCB) device, layer-parallel direct shear tester (LPDS), and the modified version at the University by Leutner. The device can be used to evaluate the performance of multi-layered bituminous systems in laboratories [77].

The Swiss Federal Laboratories for Materials Testing and Research proposed an LPDS test device as an adjustment to the Leutner test system [2]. The device allows the direct testing of samples with a diameter of approximately 150 mm and prismatic specimens with a height and width of 130 and 150 mm, respectively [87]. For the characterisation

of cylindrical samples, a direct shear device referred to as the Louisiana interlayer shear strength tester (LISST) was developed [88]. The device has two major elements: shearing and reaction frames, which are the moving and stationary parts, respectively. During the test, the cylindrical specimen is positioned within the shearing and reaction frames and locked with collars before a load is applied to the shearing frame. A gradual increase in the vertical load causes shear failure at the interface. Temperature control is achieved using a material testing system. The load force of double-layered samples (nominal diameter = 150 mm) extracted from pavements or prepared in laboratories can be measured using a fixed displacement rate of 50.8 mm/min [89]. Table 8 presents the Pure direct shear test methods available.

**Table 8.** Pure direct shear test methods.

| Categories | Device Name | Reference |
|---|---|---|
| Pure direct shear tests | Leutner shear device<br>Modified Leutner shear device<br>Layer parallel direct shear (LPDS.)<br>Louisiana interlayer shear strength (LISST.)<br>Material testing system (M.T.S.) | Canestrari, F., et al., Uzan et al. [2,73,77]<br>Ragni, D., et al. [77]<br>D'Andrea, et al. [88]. Eshed.<br>Tozzo, et al. [86]<br>Partl, M.N., et al., [89] |

6.4.2. Direct Shear Tests with Applied Loading

Loading application is a critical factor that must be considered when simulating loading conditions and when evaluating the interface shear fatigue behaviour [86]. A few shear testing machines have been used in dynamic mode due to the complexity of installation and configuration compared to monotonic modalities [88]. The double shear tester (DST) device developed by Khajeh Hosseini is an inexpensive and straightforward method compatible with universal testing machines with the provision of repeated static and dynamic axial loads in a temperature-regulated setting. DST is a reliable and reproducible method for measuring the phase angle and dynamic shear modulus in a load frequency range of 0.5–10 Hz [90]. The device can be used for unidirectional monotonic static shear tests or unidirectional cyclic fatigue shear tests. The shear stress was calculated using Equation (8) [73].

$$\tau = \frac{F\ |}{2 \cdot a \cdot b} \tag{8}$$

where $\tau$ is the shear stress (MPa), F is the shear force (N), and a and b are the width and height of the specimen (mm), respectively.

A modification of DST was proposed [91] for the study of mode II fatigue and reflective cracking performance of GlasGrid-reinforced asphalt concrete under repeated loading. Subsequently, modifications were performed at various stages to achieve the desired features, such as repeatability and stiffness. Cristina et al. used the Sapienza direct shear testing machine (SDSTM) to study the fatigue performance of the interface between the asphalt layers. The SDSTM can test double-layer cylindrical specimens of approximately 100 mm diameter and 1 cm air gap. The device comprises an LVDT for the measurement of interface displacement. Moreover, the behaviour of the interface shear fatigue can also be evaluated using the device under dynamic conditions; the loading machine has a maximum vertical capacity of 100 kN with load frequencies of approximately 5 Hz. The nature of the control system facilitates the use of any load profile [86].

A Superpave shear tester (SST) was proposed as a servo-hydraulic machine capable of applying controlled vertical and horizontal loads. The maximum shear stress was obtained by applying a constant load rate of 222.5 N/min until failure. However, stress-controlled tests were not appropriate for investigating post-failure behaviour at the interface. [92]. Consequently, a new method for determining the shear properties of asphalt mixtures referred to as the uniaxial shear tester (UST), was designed by Josef et al., wherein the shear load could be applied using either a servo-hydraulic or pneumatic press. The tester is a simple and cost-efficient alternative to an SST device with great potential as it allows tests

on as-built pavements for evaluating rutting susceptibility and lab-based Evaluation of the shear properties of asphalt mixtures [93]. The Sapienza University of Rome developed the Sapienza inclined shear test machine (SISTM), which works at a constant displacement rate under static conditions [77]. The Polytechnic University of Marche, Italy developed an ASTRA tester as a direct shear box that resembles the device mostly used in soil mechanics; however, the tester varied significantly from that developed by Uzan et al. [2]. The ASTRA device has two transducers (LVDT) to measure the shear ($x$) and normal ($z$) displacements of the specimen. The shear stress was computed using Equation (9).

$$\tau = \frac{F}{A_{eff}} \tag{9}$$

where $\tau$ is the shear stress (MPa), F is the shear force (N), and $A_{eff}$ is the effective cross-sectional area at every instant.

Double-layer cylindrical specimens with a diameter of 100 mm were tested using SDSTM. The specimen was placed between two moulds with a 1 cm gap between the two restraints. The interface was positioned in the middle, thus leaving a gap of 0.5 cm from the edge of each mould. Brown and Broderick of the University of Nottingham developed a dynamic shear box for testing reinforced asphalts, which was restricted to pure quasi-static mechanical testing at maximum forces in the range of 200–500 kN [89]. A new shear box device GS-1000 was developed by Heinz et al. The normal force in this method ranged from 0 to +1000 kN, and the shear force ranged from 200 (for tension) to +800 kN for hydro-mechanical coupled testing with a fluid pressure of approximately 10 Mpa. Figure 6 shows the ASTRA test device with the shear box device GS-1000 [94].

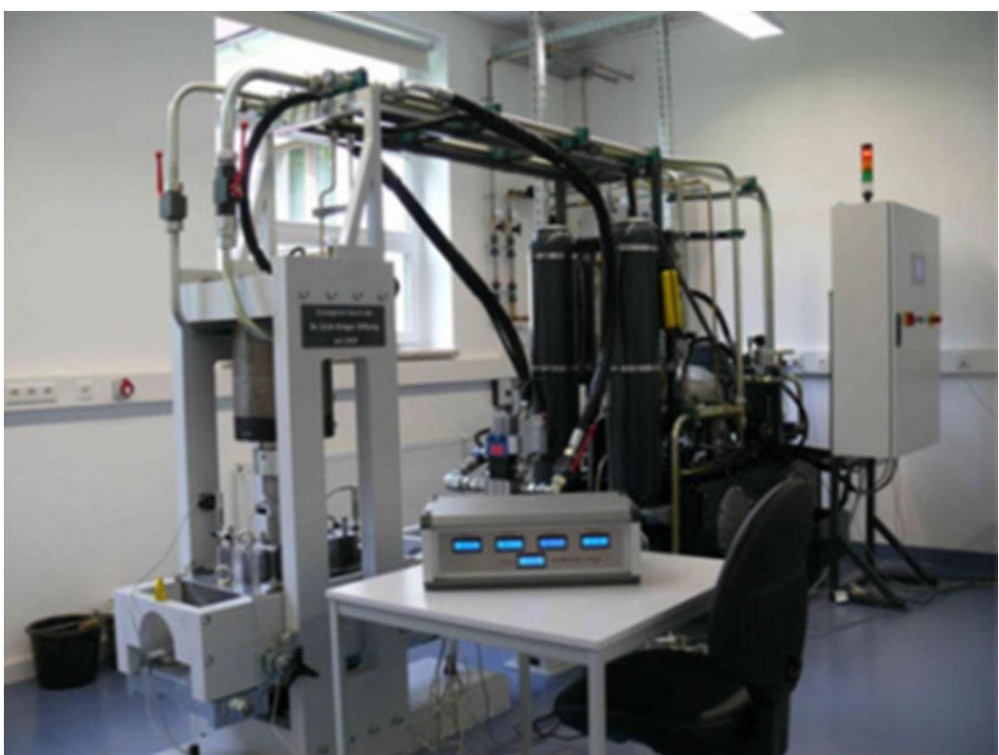

**Figure 6.** Astra test device: shear box device gs-1000.

In the Virginia shear fatigue test, the test device enables the use of repeated loads on an interlayer held between PCC and HMA. The test apparatus processes the required number of shear-load phases to failure at the interface. The repeated loading approach mimics the movement of vehicles on pavements and determines the optimal application

rate for tack coat materials. The test requires the application of a cyclic shear load of 0.1 s, a half-sine wave with 0.4 mm deflection, followed by a 0.9 s relaxation period [95].

The Illinois Centre for Transportation (ICT) designed a direct shear test apparatus for the application of shear force in the vertical direction, while normal pressure was applied in the horizontal direction. U-shaped loading eliminated the effects of bending moment caused by the eccentricity of shear force. Further, a monotonic loading mode was used to evaluate the ISS at a constant displacement rate of 12 mm/min (0.47 in/min). The device can be used to evaluate the performance of tack coats in two different modes: the cyclic mode (performance evaluation is based on the number of cycles to failure) and the monotonic mode (the strength of the tack coat is considered based on the peak load before failure) [96].

The TU-Delft four-point shear test was developed to study creep properties and shear failure in cement concrete, metals, rocks, and composite materials. The test was based on the concept of creating a vertical plane with a zero bending moment and a highly concentrated shear force in the specimen. Scholars at the University of Illinois at Urbana Champaign later developed the mode II fracture test as an extension of the TU-Delft four-point shear test. The new form captures mode II fracture work from the point of crack initiation [95].

Wheat et al. developed the King Saud University test device for bond strength testing at an asphalt interface based on the dynamic shear reaction modulus. During the test, a vertical pulsating force was applied to the interface from the actuator at different frequencies (25, 10, 5, 1, 0.5, and 0.1 Hz). The specimens were evaluated at two angles: 20° and 30°. Dynamic shear modulus tests have been recommended for determining the optimum rate of tack coat application rather than for shear strength determination [97]. Further, the Dresden dynamic shear tester (DDST) was developed by researchers at the Technische Universita, Dresden, for both quality control and interlayer bond characterisation at varying stress levels, temperatures, and frequencies. The group experimentally demonstrated that the temperature, normal load, and frequencies affect the shear stiffness at the interlayer zone. However, the adhesion of the tack coat did not contribute to layer bonding at elevated temperatures. Hence, shear stiffness was only achieved at high temperatures and with no axial load through aggregate interlocking, while upon the application of normal pressure, shear stiffness was achieved through friction between the surfaces of asphalt layers [98].

Several researchers have recommended shear tests as the most adopted method for verifying the bond strengths of interlayers owing to the similarity of the shear mode with the real cases of slippage and debonding; furthermore, they are easy to perform. Other test devices have been developed for various purposes. The intensification of research efforts has resulted in the development of various protocols for the estimation of the shear strength between pavement layers, including the ASTRA device, which complies with the European and Italian standards (UNI/TS 11214 (2007)) [99]. Further, the direct tension pull-off test (ASTM D 7522 /D 7522 M-15) is specified in the British Standard (BS EN 13863-2) [100,101], while the LPDS tester was incorporated into the Swiss Standard SN 671961. However, tensile tests should not be ignored because devices that rely on this mechanism are usually portable and can be applied on-site, particularly when testing tack coats for quality before the initiation of paving. Table 9. summarises the mechanical shear tests with loading applied, while Table 10. presents the difference between interlayer shears tests.

**Table 9.** Summarises the Mechanical shear tests with loading applied.

| Categories | Device Name | Reference |
|---|---|---|
| Shear tests with loading applied | Double Shear Tester (DST)<br>Sapienza Direct Shear Testing Machine (SDSTM.)<br>Dynamic Shear Box<br>shear box device, GS-1000<br>Virginia shear fatigue test<br>Illinois Centre for Transportation (ICT.)<br>The University of Illinois at Urbana-Champaign (UIUC)<br>The TU-Delft four-point shear test<br>(KSU) the bond strength test device<br>Dresden Dynamic Shear Tester (DDST.) | Khajeh Hosseini, M. [90]<br>Tozzo et al. [86]<br>Partl, Bahia [89]<br>Konieczny et al. [94]<br>Cho [95]<br>Leng, Ozer [96]<br>Cho and Song Hwan [95]<br>Wheat and Maurice [97]<br>Wheat and Maurice [100]<br>Leischner et al. [98] |

**Table 10.** Difference between interlayer shears tests.

| Type | Benefit | Limitation |
|---|---|---|
| Torque test | Conducted on-site or in the laboratory. Less destructive prototype field test device. | Applicable only to the top-most interface of the pavement, incorrect torque rate due to manual operation. |
| Tensile test | Conducted in existing and other laboratories, simple tests, and the study of bond strength quality of tack coats | Not feasibility at higher interlayer bonding resistance |
| Wedge splitting | They are conducted in laboratories. | Quantitative bonding problems can only determine the stress intensity factor. |
| Shear test | Simple procedure, near to on-site interlayer damage, international agreement on a test method. | Difficult to generate the pure shear stress at the interface. It needs to use more parameters |
| Pure direct shear test | Conducted on-site or in the laboratory, displacement rate of 50.8 mm/min for 150 mm diameter sample, simple test, shear force as a function of the applied displacement. | Loading conditions are not simulated, only static load application. |
| Direct shear with applied loading | Conducted on-site or in the laboratory, simulating loading conditions and shear force dynamic loading. | Understudy. Development is costly compared to other tests. |

## 7. Conclusions and Recommendation

This review aims to provide a wide-ranging overview of the shear properties of interface zones and the influence of certain factors on interlayer bonding performance. In addition, various mechanical tests are discussed. Several studies have investigated the problem of poor bonding between pavement layers and highlighted certain points of disagreement in test methods [73]. Depending on the country, several factors, such as the traffic loads, chemical properties of aggregates, and climate, affect the rheological properties of bitumen used in the tack coat layer. Another essential factor is the type of asphalt mixture and aggregate gradation, which renders the final shape of the surface texture and air voids. In addition, a discrepancy exists between the field and laboratory compaction methods used. Based on the findings of this review, the Marshall hammer is concluded to be the best method for determining the physical properties of mixtures. The Superpave gyratory compacter is the widely used device in preparing double-layer specimens, and roller compactors are recommended for SMA.

Several researchers have reached a consensus on shear tests to determine the shear strength at the interface zone because a slippery model in the debonding of pavement can be simulated. Further, the tests have been simplified further by experimental programs. Certain agreements have been added to the AASHTO and European standards with recommendations to further study the variable until greater agreements are reached to increase the bonding strength between the paving layers. The discussion in this paper provides a review of the shear properties of pavement layers to achieve an acceptable standard procedure for a comprehensive understanding of shear properties and Evaluation of interface bonding. Thus, the following conclusions can be drawn.

1. Linear relationships exist between certain factors, such as emulsion type, temperature, and application rate. An epoxy asphalt tack coat is recommended for application between the steel deck and overlay. Further, tracking problems can be solved using trackless emulsions, as has been recommended in various studies. Furthermore, the problems associated with hot and cutback asphalt binders limit their usage; consequently, certain studies have modified cutback asphalts with appropriate additives to make them environmentally friendly.

2. Fibres and nanofibers are among the newest additives for emulsions, where modified emulsions should only be applied on a clean and dry surface.

3. The type of surface, age, and texture depth have been considered by many researchers to determine the application rate; milled and older surfaces have also been observed to yield higher shear strengths.

4. A disagreement exists between the compaction methods used in the laboratory and fields owing to the different types of mechanical pressures applied. Therefore, the issues of a difference in the air void content, the distribution of aggregate, and surface configuration, particularly in SMA, are dependent on overloading.

5. The shear test is the most used method for verifying the interlayer bonding strengths owing to the similarity of the shear mode with confirmed cases of slippage and debonding. Moreover, it can be conducted with ease.

6. The weak bonding between flexible pavement layers can be described by the strength (of the adhesion and interlocking) between the surfaces, the chemical properties of bitumen and tack coat materials, and the physical properties of the combined aggregate via the ratio of fine and coarse materials to the mixture.

However, no study on the relationship between the shear properties in the bitumen of the mixture and the shear strength in the interface area has been conducted. In addition, the limitation of the gradient in one type of mixture and its Effect on the bonding strength has not been studied. This research is currently being conducted by the authors of this study, and the experimental work and results will be presented in future works. Since no clear information exists to study these variables, this study recommends continued research endeavours to analyse debonding in multi-layer asphalt pavements.

**Author Contributions:** All authors confirm the responsibility for the following: study conception and design, data collection, analysis and interpretation of results, and manuscript preparation. All authors have read and agreed to the published version of the manuscript.

**Funding:** This research received no external funding.

**Data Availability Statement:** The authors confirm that the data supporting the findings of this study are available within the article https://drive.google.com/drive/folders/1U3oaCUunTs58nHkP4LtQM1uLDGw1vMU-?usp=share_link (accessed on 11 November 2022).

**Conflicts of Interest:** The authors declare no conflict of interest.

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
