# Peer review of "Review: Shear Properties and Various Mechanical Tests in the Interface Zone of Asphalt Layers"

_infrastructures, doi:10.3390/infrastructures8030048_

Round 1
Reviewer 1 Report
1. Extensive studies about layer bonding performance have been conducted these years. As a review paper, the authors should summarize the literature properly. There are some basic mechanical analysis methods, including the Mohr-Coulomb theory, fracture mechanics, etc.
Some literatures are listed below:
[1]Hu X, Walubita LF. Effects of layer interfacial bonding conditions on the mechanistic responses in asphalt pavements. Journal of Transportation Engineering. 2011; 137(1): 28-36.
[2] Ozer H, Al-Qadi IL, Wang H, Leng Z. Characterisation of interface bonding between hot-mix asphalt overlay and concrete pavements: modelling and in-situ response to accelerated loading. International Journal of Pavement Engineering. 2012; 13(2): 181-96.
[3] Canestrari F, Ferrotti G, Lu X, Millien A, Partl MN, Petit C, et al. Mechanical testing of interlayer bonding in asphalt pavements. Advances in interlaboratory testing and evaluation of bituminous materials: Springer; 2013. p. 303-60.
[4] Tozzo C, D’Andrea A, Al-Qadi IL. Dilatancy in the analysis of interlayer cyclic shear test results. Journal of Materials in Civil Engineering. 2016; 28(12): 04016171.
[5] Song W, Xu F, Wu H, Xu Z. Laboratory investigation of the bonding performance between open-graded friction course and underlying layer. Engineering Fracture Mechanics. 2022; 265: 108314.
[6] Song W, X Shu, B Huang, M Woods. Laboratory investigation of interlayer shear fatigue performance between open-graded friction course and underlying layer. Construction and Building Materials. 2016, 115:381-389.
2. Figure 1 is not yours, you should cite the reference.
Author Response
The authors thank you for the recommendation, which was worded as much as possible to correct.

Reviewer 2 Report
The review investigates the shear properties determined with various mechanical tests at the interface zone of asphalt layers. The paper lacks cohesiveness: for instance, Eq. (1) refers to one of the possible options for simulating the shear sliding. Why this is the only one referred to? The explanation of the stiffness is now adequate, etc.
Moreover, the methodology of the review in paragraph 2 if not totally irrelevant is not correct; the last five-years internet-based search cannot be established and justified. You are strongly encouraged to have a global and wider search and examine the contribution of each single paper to the phenomenon you want to study and then, include only those publications which have made a real difference.
Figure 1 presents some data. Are these from original research of the authors or from literature?
Eq. 4 and its explanation are not correct.
A good English review is necessary and various typos which appear here and there need correction. Some citations are missing (see line 219).
Author Response

(The authors gave the same response as above.)

Round 2
Reviewer 1 Report
Authors addressed the reviewers' comments clearly. The paper can be accepted in the current form.
Author Response
Dear Reviewer 1,
The authors thank you for handling the paper and getting it reviewed. We also thank the Reviewers for their valuable work, which helped us to improve the manuscript.
Reviewer 2 Report
The manuscript has been improved but still needs some work. A careful English proofreading and editing of typos, syntax, missing punctuation, missing references (see for example p.5, l.212 & l.217 & l.266) etc. is necessary.
Moreover, there are phrases like “In the early mid-1978s” which do not make sense.
Please improve English to make it more readable.
Specific comments:
1. P.2, l.50-53: a number of models are referred to one of which is further analysed. Justify your selection. [Same comment as in review round 1]
2. Words superfluous like p.4, l.154 should be deleted.
3. In p.6, l.248-252 a research programme is mentioned. Better cite the researcher(s) who carried it out.
4. Table 2 & 3: citations missing.
5. L.397: reference missing.
6. Figure 3: reference missing.
7. Table 9-10: numbering error.
Author Response
Thank you for your helpful comments. We have revised our paper accordingly and feel that your comments helped clarify and improve our paper. Please find our response to the reviewer’s specific comments attached file.
